# “They Tarred Me with the Same Brush”: Navigating Stigma in the Context of Child Removal

**DOI:** 10.3390/ijerph20126162

**Published:** 2023-06-17

**Authors:** Joanne McGrath, Monique Lhussier, Stephen Crossley, Natalie Forster

**Affiliations:** 1Department of Social Work, Education and Community Wellbeing, Northumbria University, Newcastle upon Tyne NE1 8ST, UK; monique.lhussier@northumbria.ac.uk (M.L.); natalie2.forster@northumbria.ac.uk (N.F.); 2Department of Sociology, Durham University, Durham DH1 3HN, UK; stephen.j.crossley@durham.ac.uk

**Keywords:** stigma, social harm, mothers, child removal, health and social inequality, multiple disadvantage

## Abstract

Child removals are increasing in England and Wales. Family court involvement is particularly common among women with multiple disadvantages, and the rates are higher in economically marginalised areas. This article aims to explore women’s narratives of child removal within life stories of homelessness and examines how stigma, power and State surveillance manifest in their experiences. Data drawn from qualitative interviews with 14 mothers in the north-east of England who had experienced the removal of their children through the family courts are explored within the wider context of a neoliberal political agenda of “troubled families”, and in particular, “deviant mothers”. The participants describe how stigma structured their interactions with social services. Despite the known poor outcomes associated with child removal for both mothers and children, professional involvement often tapers off afterwards, with little support for mothers. Drawing on women’s accounts, we seek to illuminate their experiences of child removal and enhance our understanding of how stigma plays out in statutory settings, further entrenching social exclusion and ultimately increasing health inequalities.

## 1. Introduction

Over the past two decades, the number of children subject to care proceedings in England and Wales has increased significantly [1]. The impact on marginalised populations has been disproportionate, with significantly higher rates of child removal in economically disadvantaged areas [2]. The north-east of England has the highest rate of referrals to children’s social care, with a 77% increase in its care population since 2009 [3]. In response to this crisis, a report by the Independent Review of Children’s Social Care recommended wide-ranging reforms, amounting to a “radical reset” of a system found to be both inconsistent and “increasingly skewed to crisis intervention, with outcomes for children that continue to be unacceptably poor” [4]. Marginalised mothers facing multiple adversaries of poverty, trauma and substance-use are particularly at risk of child removal [5]. One of the criticisms of the family court system is their focus on these individual pathologies—the so-called “toxic trio” of poor parental mental health, substance use and domestic abuse [6]—which has been criticised for being “deeply stigmatising” [7], as well as for being overly simplistic and not sufficiently taking inequalities and contextual factors, such as area deprivation, into account [8].

For mothers with multiple marginalities, the cumulative effect of these factors can be described by the term “intersectional stigma” [9], which may present additional barriers, thus undermining their mothering goals [10]. Stigma is defined as a social construct whereby an individual is discredited through the influence of components of stereotyping, prejudice and/or discrimination [11]. Goffman’s theory of “spoiled identity” [12] describes how people are stigmatised in relation to others; thus, stigma is linked with both intersectionality and power [13]. Motherhood is a particularly potent vessel as a gendered form of stigma, which can be “constructed and reproduced locally through various pathways” (p. 625, [14]). The concept of idealised motherhood as an expression of ideal femininity is grounded in patriarchal and neoliberal discourse [15,16]. Mothers stigmatised by “otherness” include those singled out on the basis of poverty, race, care experience, substance-use, age, domestic abuse and criminal justice involvement [17,18,19].

The impact of such trauma and adversity often leads to these women being additionally stigmatised as “unfit” mothers [18], being seen as “hard to reach” and falling through gaps in support. Women involved in recurrent proceedings are often young when they have their first child, with patterns of adverse experiences in childhood and adulthood and histories of involvement with the care system themselves [20]. Previous findings highlight that stigma and the fear of loss of custody are important factors that prevent mothers who use drugs from accessing important health and social support, thereby increasing their isolation [21]. Stigma manifests in these mothers’ lives through increased scrutiny and judgment, with their experiences of motherhood being discounted in favour of what is typically considered superior or “expert” knowledge [22]. Minaker summarises how marginalised individuals are thus often “placed in the space of ‘other’, puzzlingly unseen but hyper-visible” (p. 2, [23]).

The parenting surveillance that takes place in social service settings has been noted to disproportionally sanction marginalised mothers [24], where increased scrutiny and additional “protection” of their children is justified due to them being deemed “threatening” or “unruly” [25]. Within the construction of women as being primarily responsible for child welfare and safeguarding, reinforced by gendered discourses of parenting [26], the focus on victims of domestic abuse (i.e., the mother) rather than the perpetrator leads to mothers being viewed as “unprotective” [27]. The surveillance of victims of domestic abuse is thus justified by blaming them for “failure” to prevent the violence and being unable to protect their children from an abusive home [28]. As McDonald-Harker (p. 324, [29]) argues, “…[I] n contemporary neo-liberal times the “abused woman” is responsibilised, pathologised, and her plight decontextualised from systemic factors like gendered violence and economic marginalisation that constrain her choices…. The ways non-criminal justice agencies perceive and treat mothers who grapple with domestic abuse situations are in keeping with a wider culture that demonises mothers.”

If deviance and stigma can be effectively conflated, then the austerity of neoliberalism during the ‘hostile decade’ of austerity in the UK can be seen as a particularly punitive manifestation of state power being exercised through societal institutions [30]. Welfare stigma is not a new concept [31,32,33,34] and, historically, state interference in women’s reproductive lives is associated with mothers who are marginalised in society, and often labelled as “deviant” [17]. The origin of modern day child protection and safeguarding was in the 19th century child rescue movement, which provided institutional care and support for the poor, destitute and orphaned young. This was influenced by the government welfare policy as set out in the 1834 Poor Law [35] that focused on separating those “deserving” of help from the “undeserving”. The “undeserving” were a focus of the Conservative-led governments decade of welfare cuts, recently compared by the UN to “the nineteenth century workhouse” [36], which was justified by state-crafted stigma, and which has had “disproportionate impacts […] on those already on the losing end of the British class society” [37].

This rapid shift, alluded to by Jones [38], has seen stigma (norms marking an ontological deficit, non-conformance or shame) being redefined as deviance (norms marking a moral deficit, non-compliance or blame), ‘skewing’ social norms of shame and blame, which Scambler [33] terms as a “weaponising of stigma”. Thus, stigma operates as a powerful tool to police and regulate the most marginalised [39]. Lone mothers, in particular, have been branded “failed neoliberal subjects” (p. 232, [40]), as vessels of unregulated and amoral female sexuality and therefore a threat to society [40]. Poverty is the most pervasive factor associated with child protection involvement [41], the influence of which works directly and indirectly (through parental stress and neighbourhood conditions), with contributing factors such as mental health problems, financial difficulties and a lack of social support [42]. The recasting of vulnerable people as “scroungers” was typical of the Troubled Families Programme whereby, shifting attention from the failures of the state, neoliberal policies located problems within deviant families (more specifically, the mother), thus ‘ultimately holding women accountable for the wellbeing of the nation and for poverty, crimes and other social ills that may threaten this’ (p. 131, [43]). The systemic punishment of “deviant mothers” through the family courts can be seen in the wider context of Wacquant’s observation of the “double dimension” of marginality, both “material and symbolic, as well as to the other state programmes that purport to regulate “problem” populations and territories’ [44].

In this sense, child welfare represents a form of structural stigma, which includes the societal conditions, cultural norms and policies that constrain the well-being of stigmatised groups. Child removal has been found to exacerbate marginalising issues and is “firmly the gateway to further adversities” [45]. We hypothesise that stigma is one of the conduits of this gateway. Link and Phelan [13] note that one function of stigma is to limit access to resources that support wellbeing, thus maintaining the unequal distributions of power [46]. In the context of intersectional marginalisation, the potential loss of housing and welfare benefits has important material and symbolic value, thereby potentially exposing people with marginalised and stigmatised identities to ongoing stigmatisation [47]. The psychological consequences of stigma are profound and may compound the known damaging effects of child removal on both mother and child that have already been well documented, with the trauma of child removal in the absence of support exacerbating risk in other aspects of women’s lives [45,48], including an increased risk of suicide [49].

The mechanisms of stigma relating to child removal are not yet well understood; however, in relation to other types of gendered stigma, a three-domain framework has been suggested [50], which we have adapted here: first, perceived stigma, whereby a woman’s awareness of the devaluing attitudes of others regarding child removal, which leads to the expectation of discrimination as a result. Second, internalised stigma, which results from internalising devaluing social norms, attitudes and beliefs relating to child removal, perpetuating guilt, shame and other negative feelings. Finally, enacted stigma describes the actual experiences of discrimination or negative treatment from others directly related to child removal.

Given that women often “fall through the gaps” in policy and practice following child removal, we explore the intersection of poverty, stigma and gender that affects individuals who have lost custody of their children through the family court system in the UK, as described through their recollections. Specifically, we aimed to clarify the unique elements, attributes and features of the different types of stigma experienced and its potential relationship to how stigma plays out in this context. We hope that this greater understanding will encourage more protective working practices, thus avoiding unnecessary child removals and the perpetuation of the cycle.

## 2. Methods

The findings presented here are drawn from a wider ongoing examination of the experiences of women with co-occurring issues, including chronic homelessness, substance-use, poor mental health and domestic abuse, focusing on the impact of this on their health and access to services. A total of 20 women took part in semi-structured life-course interviews, which were conducted in the north of England between October 2021 and February 2022.

The inclusion criteria for participation was identifying as female, being over 18 and having experience of chronic homelessness (more than 3 separate instances or longer than 3 months), poor mental health and substance-use and/or domestic abuse, as well as being single (defined as not currently having dependent children). Participants were identified and invited to take part in the study by gatekeepers from two community organisations or recruited via opportunity sampling. Where appropriate, staff with prior knowledge and understanding of the women facilitated in-person introductions to the researcher, who briefly introduced the study, and then screened potential participants for inclusion. One participant was recruited directly from the street, where she was rough sleeping. The participants were interviewed at each location until data saturation was reached. The interviews lasted between 25 and 140 min and were transcribed verbatim. The semi-structured nature of the interviews allowed the lead researcher to probe on areas of interest relating to health, but also allowed an opportunity to move beyond the predetermined questions by allowing the participants to narrate their stories in their own words, focusing on what they felt were important experiences. Participants spoke spontaneously, openly and often at length about child removal, which generated significant amounts of data. Ethical approval was obtained from Northumbria University prior to data collection. Names have been changed, and exact locations are not described in order to preserve the anonymity of participants. Every participant was given a voucher once the data collection was completed to thank them for their time and insights.

## 3. Analysis

As the project was not originally focused on child removal, the data were inductively interrogated on a case-by-case basis to maintain engagement with the original account before searching for common themes across interviews [45]. The initial coding was informed by the existing literature, but also openness to how women described their accounts of loss. We began by ascribing descriptive codes to the initial cases to identify commonalities and develop explanations. Research notes made by the lead researcher and debriefs between the lead researcher and her primary supervisor helped to identify and make sense of the experiences, thoughts and feelings during the interviews, which were often emotive and carried a heavy emotional load [51]. Discussions then took place between the authors to compare their interpretations of the data and confirm the final themes. A verification workshop was organised towards the end of the data collection period, which ten of the participants attended, serving as a reference group [52]. They were presented with preliminary findings and shared further insights. Child removal emerged as a consistent and unexpected finding and a salient experience shared by many of the women. Data pertaining directly to child removal, identified in 14 of the participants’ accounts, were extracted for further analysis, and are presented in this article.

## 4. Findings

### 4.1. Perceived Stigma: Making Sense of Spoiled Motherhood

The literature on the maternal identity of substance using mothers has shown that, for many women, motherhood is seen as a step towards a positive, legitimate social identity [53]. Most of the women reflected positively on their children’s early years as a period of normality and a chance to embrace a new start: *it was fine for a full 8 year. I wasn’t taking nothing, I wasn’t doing nowt. I was just high on life! I had always wanted a bairn*. (Dee)

Studies have shown that despite the lifestyle of mothers who use substances typically being seen as outside of social norms, these women tend to have traditional values about pregnancy and motherhood [19]. Thus, some of the women’s narratives seek to demonstrate their fitness as mothers:


*I got pregnant, had her, kind of kept hold of him [partner] cos I believe a family should be a family. Till the age of 5 brought her up and that… we were fine. We were a good little family to be honest, like we had everything.*
(Gillian)

Similar findings have emerged from other studies of marginalised mothers. Enos’ study of mothers in prison [54] found that they sought to emphasise their morality, and as Enos describes it, their “claims” to motherhood on grounds such as their biological and emotional bonds with their children. This could be one way to interpret Michelle’s recollection of her daughter’s stay in hospital following serious illness, emphasising her devotion to her care: *I was in the hospital the whole time sitting on a blue chair for weeks, no sleep, wouldn’t leave her. Wouldn’t walk, I’d never leave me bairn*.

Many of the women’s accounts reflected a desire to demonstrate their mothering in terms that aligned with societal expectations: *“I mean fair enough, don’t get us wrong like I mean you’ve got people who beat their kids up and stuff—but there’s never been anything to do with that. But the kids have always been clean, tidy, at school on time, everything”*. (Rosa)

The mothers’ accounts of loss are highly conscious of their stigmatised identity and try to distance themselves from other mothers who are perceived as less deserving, whilst striving to explain their maternal fitness.


*It just seems like nobody listens sometimes…It was just totally wrong. Like I totally agree with like “you can’t look after the kids” or if you can’t look after the kids or there’s different situations but when you’re actually literally no, that never should have happened. It’s not just affected me it’s affected me kids. Like that should never have happened. Never at all. And everything’s whipped away from me and the kids. So it’s just horrible.*
(Sienna)

Some of the other women’s accounts indicate individual stigma management. For example, Sharon was in a relatively powerful position compared to the other participants as her children were older and shared a well-established relationship with her that she was able to leverage to ensure that she was still able to see them. Indeed, they chose to keep running away from care, returning to her, a point that she referred to with evident pride several times during the interview:


*And all he does is run away from care, come back to mine they knew where to pick him up cos I was letting him in. I don’t care what they say, he’s my son.*


There was likely an element of performing their identities during the interviews, working against or casting off elements of their stigmatised identities during the interviews. Suzy repeatedly emphasised her position as a mother, describing her encounters with social services in terms of power and resistance:


*Social Worker knocks on the door and he went … I’m just checking to see if [child] has come to live with you blah blah. But he seen a can on the fireplace, so I said, I drink and anyways you know I drink. So erm, he come back again right, he knocked on me door—I wasn’t in. I was not in. It’s a downstairs flat right, so he come back again… And he was the boss you know. I told him to fuck off. The boss of social services… No cos I’m cheeky. I’ve to be wide, I’ve got to be wide [wide awake, sharp witted] like that, cos you know what it is, they would have us [take advantage]. And nah. Nah. No way.*


Suzie’s defiance and refusal of a victim identity is echoed by the literature describing how women manage a damaged sense of self when stigma is internalised, meaning that the woman in question accepts stigmatising social norms whilst maintaining and managing a damaged reputation [55,56].

### 4.2. Internalised Stigma, Shame and Silence

Research demonstrates that being a victim of domestic abuse has a significant impact on the ability of women to protect their children [57], who can feel a loss of control over their parenting abilities and resulting feelings of guilt, self-failure and blame [58]. The women’s narratives of domestic abuse revealed a direct correlation with their children being taken into care*—You were with your partner but he was violent so you weren’t allowed to keep your child* (Kassie)—but were also heavily influenced by internalised stigma and shame.

Michelle was a victim of domestic abuse at the hands of an ex-partner. Despite acknowledging her powerlessness to escape a violent relationship, she associated victimhood with choice, internalising those narratives of individual blame and linking this to being unable to meet the standards of good womanhood:


*Erm, I’m a nightmare, I’m a disgrace. I hold me hands up I’m a fucking disgrace. I really am, as a parent I’m terrible. I’m a disgrace and it’s all me own fault. It’s all me own fault. Nobody else’s, me own. I made the decisions, so I have to take the consequences of it.*
(Michelle)

Michelle’s statement that she was a “disgrace” was loaded with the external judgement that society places on these women. She loved her children and was devastated by their loss; however, her trauma was compounded by feelings of shame.

Stigma by association describes the process in which a person is stigmatised by virtue of association with another stigmatised person [12]. Sally talked about how she came to the attention of social services due to her husband’s offending; despite her being proven to have no knowledge of, she was judged to be unable to adequately protect her children:


*Unknown to me his family had a history of abuse as well. So social services targeted that … my solicitor made clear there was no way it was my fault; it was his. And they tarred me with the same brush that’s the only way I can say it, cos he’s like that it’s your fault you’re classed as the same”.*
(Sally)

The women described being judged harshly by being associated with perpetrators of crimes that conflict with the ideals of womanhood, even when they themselves were not involved. In Leona’s case, her feelings of internalised shame at her failure to live up to societal standards of motherhood led to an ongoing impact on her mental health and withdrawal from relationships: *I think I just spiraled too … I think I thought the world was against us and, punishing myself as well … the kids, being on methadone, I just, that’s all I think about*. (Leona)

### 4.3. Enacted Stigma

The participants reported being devalued, rejected or treated unfairly during care proceedings. These forms of received stigma [59] were most often described in relation to the participants’ experiences of interacting with social work agencies, which were described as overly bureaucratic, punitive and stigmatising [60].

Tyler’s [34] observations about the way shame “lives on the eyelids” provides some context for the women’s experiences with services, with stigma shaping the ways in which we see and are seen, and noting that an awareness of the likelihood of being judged could contribute to women’s isolation, making it more difficult for them to access support:


*The kids were took into care… I was made to feel about that big. And it’s not very nice… I didn’t ever want to be made to feel like that again, and I never did. I hated social services. I had to keep away from them cos they made us feel sick.*
(Sally)

The involvement of social services was perhaps unsurprisingly universally described as a negative experience by the women who had lost their children through court proceedings. The use and abuse of power within child protection services has been reported previously, through threats, silencing and coercion [61]. Being pre-judged as a product of the care system also influenced the women’s narratives of the stigma attached to previous social services involvement: *[social workers] were just “your mam was an alcoholic. You’re the same”. Basically just tarred me with a brush*. (Rosa)

Rosa has a diagnosed learning difficulty and Autistic Spectrum Disorder (ASD) and is care experienced. She describes how an interaction with a psychiatrist in family court left her feeling baffled and disempowered:


*They had a psychiatrist who came into court and because I’d never heard of the saying “a bird in the hand is worth 2 in the bush”, I’d never heard of it in my life before. And she said “what’s a bird in the hand worth?” I went “I haven’t got a clue, I don’t know what you’re on about”. Where did that come from? What’s that got to do with me kids or anything like that? And she went, “Well can you answer the question?” I went “I’ve never heard of it so obviously clearly not”, she went “it’s worth 2 in the bush”. And they went, because of that I wasn’t capable of being a parent.*


Furthermore, the mothers felt dehumanised and belittled by practices that formed judgements about their parenting and called into question the truthfulness of their testimony. Rosa described how she discharged herself from hospital, still in pain after a serious operation, to see her son:


*They gave us the bairn and [he] was kicking at us, he was only just over a year old so he’s kicking his legs and they’re going in and I’m wincing in pain, and I had turn him round so he wasn’t digging his knees in. And when I did that, they went “you’ve got no attachment to him”… They still didn’t put it down in the report, they were saying I was lying about being in hospital, and having the operation.*
(Rosa)

The way that this episode was described sounded akin to a well-rehearsed story and speaks to the ways that the women were performing their identities throughout the interviews, making sense of their experiences, positioning themselves and possibly trying to cast off stigma during the interview process itself. The women’s accounts of their interactions with social services paint a picture of dehumanising interactions:


*I had [the baby] at quarter past 4 in the morning, the social worker was there at 11 telling us to sign a bit of paper to sign her over. I refused. So they said I couldn’t stay with her, so I walked out of the hospital erm and I was getting to see her at the hospital but the social worker rang us and said the foster woman who was there for [baby], didn’t think it was in her job description to be here when I was there … the last time I was there, 12 o clock I was there and I rang the social worker and I went where are you I’m meant to be seeing the bairn. She went I’m not coming you didn’t text us yesterday. So the nurse took us around to see the bairn in the fishtank, I wasn’t allowed to pick her up or hold her. Then I was 7 min late for one of the [appointments] at the civic. She wouldn’t let us see her then. Seven min late. I said you go home tonight and see your children and see me begging on my knees to see my baby girl, and that was the last time. She said that was the last time I’d see her. It killed us. She put her little hand on me face (cries) it breaks me heart. She shouldn’t be where she is man. I’m a lot of things but she wouldn’t want for love you know?*
(Michelle)

Michelle’s account mirrors a common experience for new mothers making memories with their newborn, recalling her daughter’s hand on her face with the added poignancy that she knows that these moments will end and she will have to say goodbye, albeit without closure.

Persuading social services of being a “good enough” mother depends, to some extent, on the mother’s ability to operate well in the world of the family courts and to develop trusting and open relationships with the social worker. Gillian contrasts her first social worker, who she felt she could trust and who subsequently judged her parenting adequate without support, with her second social worker five years later, who was more rigid and rulebound, ultimately leading to the court removal of her child:


*When they did knock on me door all them years later, they were, I got a funny one. And she was a bit of a twat. And I was scared to tell her anything, I kept stuff back from her, that’s why eh, that’s why I lost her [daughter]. I just didn’t open up to her you know what I mean?*



*[researcher]: Yeah, you didn’t trust her?*



*Nah. I didn’t like her. I just didn’t like her. Do you know what I mean, that didn’t go down very well. I loved the other one. The one that was in my life for the first 4 months I wanted him to stay. He said I don’t need to be here anymore. I didn’t want him to leave, got him a present. He was brilliant. Then I got this other one… she was out of university and stuff and she was just textbook. I’ll be honest like, aye, I should never have lost her [daughter]. I shouldn’t have lost my kid like. Nah.*
(Gillian)

Gillian’s reluctance to engage with the social worker demonstrates the importance of interpersonal skills and approach; she was perfectly happy working with one of her social workers, and he with her. Therefore, she did not reject social services involvement outright. Her description of the second social worker as “textbook” perhaps implies that this social worker had a more rigid view of what constitutes ‘good motherhood’ in accordance with the normative views of this topic in society—stigmatised identities/certain contexts are by definition and automatically seen as problematic for child welfare, and therefore do not fit the definition of textbook parenting. This social worker was less experienced, “out of university” and had perhaps therefore not yet developed the experience to apply nuance to the textbook motherhood model, and (as she says) there was a lack of trust that potentially went both ways.

### 4.4. “Everything Just Seems to Have Stopped”: Lack of Support Post Child Removal

The trauma women experience as a result of the loss of their children is often accompanied by a dramatic decline in their circumstances: *I ended up in like a hostel. I lost me car, lost me job, I lost me home with everything in it, lost me kids. I lost everything from there… it really upsets us like even thinking about me kids* (Sienna). However, following the conclusion of family court proceedings, there is an almost total absence of statutory support for the mothers.

In the UK, only indirect contact (commonly termed “letterbox contact”) is allowed between mothers and children who have been adopted. This involves the exchange of letters and photographs, but this can be sporadic or stop altogether. This unique form of loss, whereby a mother has lost children and is unable to locate them or even know where they are living, was described by Morriss as “Haunted Motherhood”, a sort of purgatory state where mothers are haunted not only by shame, but also the ambiguity of their loss, “unable to follow the customary grief rituals of bereavement as their child has not died but is alive” [62].


*She’s been writing and they’ve been writing back and been in touch and everything just seems to have like stopped. And I don’t know why. And I don’t know where they are and who they’re with.*
(Carina)

The lack of social validation surrounding forms of loss that are out of place with societal norms is sometimes termed “*disenfranchised grief*” [63]; this describes a type of grief and emotional distress that is unlikely to be acknowledged or to elicit sympathy from social networks. Sally explained how many of her closest friends are unaware that she is a mother: *I remember their birthdays cos I light a candle […] not many people know they were adopted*. (Sally)

At an interpersonal level, stigmatised individuals may become socially isolated and withdraw from others to avoid discrimination. The stigma literature describes how stigma related to circumstances or experiences that can be hidden can carry additional costs associated with behaviours that manage the stigma, such as keeping the experience secret or suppressing intrusive thoughts [50]. Thus, these women are left to cope with overwhelming feelings of loss without the support of friends and family. The lack of support after child removal, at the point where mothers describe their lowest ebb, seems to be missing this crucial time when women need help:


*They kept us in the mother and baby unit for two weeks after they took me child off us. And expected us not to get off me face. When I’m hearing other people’s babies crying. That fucked with my head, completely. And then I met the partner I’m with now and had [name] and she got took off me on my 21st birthday.*
(Amy)

Interactions with services not only failed to help or address the women’s isolation, but also, as Rosa describes, compound their distress:


*I had [child] and I had me own tenancy, I had everything for her, everything was brilliant and then social services came along and pff, blew everything up. So as soon as they took her, they didn’t give us no help to, not be a parent. Cos one minute I was a full-time parent, the next minute [cries] it was just an empty house full of stuff. There’s no noise, no nothing. Just, what do I do? I’d get up, I would hear her cry on a morning, and I’d get up and realise, she’s not there. And I went to try and speak to people about it, I’d say look I need some help here, I can’t cope. I’m not coping. And they wonder why I don’t sleep and they wonder why I’ve got all these problems…They just wanted me off their case. They just left us to it. And this is how I am now. After they took [child] when she was 18 month old that’s when I hit the drugs. I just wanted to die. I just didn’t want to be here anymore. My purpose, my whole entire purpose in life was gone.*
(Rosa)

These narratives speak of a system that not only does not acknowledge women’s identities as a mother, but also denies their motherly love and is cruel and traumatising, directly influencing riskier and harmful substance using and the stigma attached to this. This lack of support from statutory services echoes the self-imposed barriers to relationships that might have provided support elsewhere or further entrench women’s place in social networks that are damaging to their health [64], thus reinforcing the incapable mother stigma.

## 5. Discussion

This study illustrates how stigma can serve to entrench and perpetuate inequalities [34]. In addition to Link and Phelan’s observation that stigma works to deny access to those resources that materially support well-being [13], our study found that mothers were stigmatised through restricted access to resources that support motherhood as a positive and socially acceptable identity. Building on previous research findings of circular patterns—whereby childhoods marked by abuse or neglect frequently evolve into adulthoods where subsequent substance use is utilised as a coping mechanism for trauma and poverty, leading to dependency on abusive partners and a lack of social support [64]—our study found that all of these factors conspire to hinder women’s ability to fulfil their mothering goals [65].

Stigma acts as a vehicle through which the problematising of the mothers in this study is expressed within austerity politics, which pathologises individuals and locates responsibility away from the state. Indeed, whilst the majority of the women in this study are products of the care system, the failure of that system to protect them is used as evidence of their unfit parenting. The tendency for women to demonstrate their fitness as mothers often presented itself in this qualitative research, characterised as a “performance of good motherhood”, which was reconceptualised by Nichols [66] as a representation of the struggle between marginalised women’s resistance to and internalisation of stigma. One of the most salient findings from our study is how mothers’ experiences of internalised stigma associated with motherhood and its loss shapes their identity and their ability to access support. The “spoiled identity” of motherhood persists, and thus structures women’s experiences. We also found that perceived stigma was closely related to the mothers’ predominant experiences of intersectional stigma; thus, motherhood represented a potentially unspoiled identity that was highly valuable.

Motherhood identity in this context is therefore framed by stigma that is both structural and *structuring* the very mechanism through which inequalities are enacted and maintained. Existing research has tended to focus on barriers to treatment and support for pregnant women and mothers with multiple exclusions, with comparatively minimal focus on how systems can address these barriers or how women’s internalised stigma shapes their interactions with services and other sources of support. One of the most important findings in this study is the predominance of enacted stigma, which was previously theorised as a feature of marginalised women’s interactions with services [50]. Whilst it is well-established that experiences of stigma can prevent people from accessing support, the women described highly discriminatory interactions with services loaded with enacted stigma, thus suggesting its prevalence.

Whilst the experiences of stigmatised mothers are placed centrally here, it is worth noting that stigma operates bi-directionally within child protection, within a risk averse system that also apportions blame to the social workers who make decisions under constrained conditions, whereby they themselves may be subject to stigma and shame: “social work as a profession has become driven by a fear of failure—ultimately a fear of being vilified in the media and publicly humiliated” [67]. Thus, the women in this study described “textbook” decisions made with little attention given to their potentialities and capabilities.

Within this constrained setting, there is a tendency to consider child and mother as independent—rather than *inter*dependent—units, with problems being circumscribed in time and space rather than understood in their historical, spatial and social contexts. The women are themselves the products of malfunctioning socio-relational-systemic systems that have failed them and are being replicated on their children. Their grief is therefore both about the loss of their children per se and about a history that repeats its mistakes and feels inescapable, reinforcing established and internalised stigmas within a time-constrained system in which family judges’ workloads are “remorseless and relentless” [68]. Whilst the protection of at-risk children must be prioritised, the consequences of treating an intersectional, gendered social problem as an issue of child welfare that is self-contained and easily circumscribable are clearly evident.

A lack of early support for families was highlighted as a salient factor in the Independent Review of Children’s Social Care report [4], and given the high risk of future harm to mothers here, there is a significant wasted opportunity for intervention. Stigma represents another barrier to being able to source this support from both services and peers. Indeed, a High Court Judge estimated that approximately one-third of removals could have been prevented if the right support had been provided [69]. Given that women typically receive little or no follow up support from public services following the removal of their children, the women’s retrospective accounts enable their experiences in the immediate and longer term to be shared. Increasing engagement with formal and informal sources of support depends, to an extent, on understanding the multitude of stigmas to which these women are subjected, with a view to reducing stigmatising interactions.

## 6. Limitations

Given that the research had not been specifically focused on child removal, there were several occasions in these extracted data where follow-up questions to expand the participants’ reflections would have been useful, but were not asked, or where discussions occurred as part of wider conversations about trauma and substance use, rather than child removal per se. The sample size, at 14 participants, was small. This may limit the depth of analysis, and the extent to which findings can be generalised elsewhere. However, the richness of the participants’ narrative reflections and the prevalence of the themes relating to mothering and stigma in the data warranted this exploration.

## 7. Conclusions

Women’s experiences of stigma reveal multiple pathways through which stigma can exacerbate existing health and social inequalities. Both internalised and enacted stigma can effectively reduce the support available to women at a particularly traumatic time in their lives. The stigma, discrimination and poorer health outcomes associated with child removal can therefore be addressed at multiple levels [70]. The current system is flawed. Women with intersectional needs and complex histories are faced with a system that is often strictly risk averse and deficit driven. Effective family court processes and policies must be informed by an awareness of the role of stigma in women’s pathways to multiple exclusions and the subsequent consequences for their parenting opportunities, and that of their children.

## Data Availability

Anonymised data sets can be accessed on request by applying to the lead author.

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
