# Peer review of "“They Tarred Me with the Same Brush”: Navigating Stigma in the Context of Child Removal"

_ijerph, 2023, doi:10.3390/ijerph20126162_

Round 1
Reviewer 1 Report
Thank you for the opportunity to review this paper. It is clear, well-written, well- referenced, and an interesting read. A lovely piece of work! I can really offer only one comment. I would ask you to consider using 'substance use' in all cases where you use substance misuse, substance abuse etc. The same applies in suggesting 'domestic violence' over 'domestic abuse'. I think the more generic terms are less stigmatizing than the others used in the paper.
Author Response
Thank you very much for the positive comments which are appreciated! We thank you for your suggestion to replace the suggested terms with “substance use”, we agree this is a less stigmatising and more appropriate term and we have updated the paper accordingly. We have standardised the use of the term “domestic abuse”, replacing “domestic violence”. We used this as the preferred term to encompass all types of domestic abuse including emotional and psychological abuse. Many thanks
Reviewer 2 Report
Thank you for this interesting article, which seems to be a reflection of a very complex social situation, in which legal and ethical aspects of child removal need some thoughts. Without evidence of child abuse, taking a child from his/her mother under the idea that the mother may not be fit for motherhood (according to the law and the prevailing stereotype of a “good mother”) could be cruel for both mother and child, but the goal of doing this is to provide the child a home that will cover all his/her needs (shelter, safety, food, clothing, education, medical care, etc.), so it is difficult to set a line because it may not be up to the mother to fulfill the child’s needs, even if she loves the child. It is a very complicated issue and each case needs specific considerations. Social workers indeed need proper training and skills to perform this task. I enjoyed reading this article.
I understand that the project was not originally focused on child removal, and that child removal emerged as a consistent finding, so testimonies from 14 out of 20 participants who suffered from child removal were used for this article. I wonder if 14 women are enough to exemplify the overall situation of child removal, and fully represent the intersection of poverty, stigma and gender in the UK that the authors are trying to dissect. Probably the authors could discuss the sample size as a potential limitation of the study?
I believe this study handles very sensitive information from the participants, and therefore the authors should provide an ethical approval number, as well as IRB and informed consent statements.
English is fine, a minor revision of the final version could be recommended.
Author Response
Thank you for the very helpful comments. We have added some text to note the limitation of the small sample size. We have added the requested ethics reference and informed consent information at line 513.
Reviewer 3 Report
The study examined women narrative of child removal within life experiences of homelessness. It looked at it from stigma, poverty and other significant areas.
The use of neoliberalism and patriarchy to explain feminism and idealised mothers in the introduction section is commendable and insightful.
However, in the "Abstract" you stated that 14 mothers from Northwest England was interviewed. But in the method section (see line 152), you stated that 20 women was interviewed in North England. Please review and align
The literature used in the "Result" section is dated and too old
Conclusion is missing in the study
contribution to knowledge is missing
recommendation, if any, is missing
Kindly simplify your tenses. Long tenses are boring
Author Response
Thank you for the comments, which are appreciated. We have stated at line 52 that the wider study interviewed 20 women, however this was not originally focused on the topic of child removal. At line 191 we note that the data from 14 women met the inclusion criteria for this study and was extracted and discussed here. Having reviewed the literature section, most of the references are recent and any older literature is directly relevant to the study. We have added a conclusion at line 504, which contains recommendations.
Reviewer 4 Report
The manuscript addresses the intersection of stigma, gender, and other “minority”-related variables (e.g., poverty) in a qualitative study that examines women’s narratives of child removal in England. The manuscript is well-structured and provides important information as to gender disparities and internalized stigma affecting some parts of the population.
As a minor revision, I noticed that there are different types of font throughout the text. Please adapt the text accordingly.
In the very first lines, the authors might consider providing some numbers (i.e., percentages %) for the rates they refer to.
The Results section is particularly well-articulated and the dialectic between the quotes (of the mothers interviewed) and the comments sounds really good. Well done!
The Discussion section is also rich and exhaustive, but a Conclusions section is missing. Therefore, I would suggest that the authors write a brief section “Conclusions” to sum up their findings.
As a typo, lines 403-413 should be italicized.
Author Response
Thank you for these helpful and constructive comments. We have added some extra figures which give context to the rates we refer to at lines 29-31.
We have added a conclusions section at line 505. Many thanks for spotting the typo! Now fixed.
Many thanks!
Reviewer 5 Report
Thank you for the interesting reading. Here are some of my minor suggestions and comments.:
- I believe that the lines 403-413 are quotes from Rosa, however this paragraph is not in italic like other quotes. Please make it more clear.
- Results: I'd suggest to use results section and sub-sections to present the findings rather than presenting and discussing them imeediately (e.g.: ll 199-202, 206-208 etc.)
- Discussion: follow the result section's flow.
Author Response
Thank you for the helpful and considered comments. We have italicised the lines at 403-413 as suggested. We have considered your suggestion to rearrange the results and discussion but other reviewers felt this was well articulated so we have decided to leave this section as it is.